# Genome-Wide Identification of the Highly Conserved *INDETERMINATE DOMAIN* (*IDD*) Zinc Finger Gene Family in Moso Bamboo (*Phyllostachys edulis*)

**DOI:** 10.3390/ijms232213952

**Published:** 2022-11-12

**Authors:** Xiaoqin Guo, Minshu Zhou, Jiaoyu Chen, Mingxia Shao, Longhai Zou, Yeqing Ying, Shenkui Liu

**Affiliations:** State Key Laboratory of Subtropical Silviculture, Zhejiang A & F University, 666 Wusu Street, Hangzhou 311300, China

**Keywords:** moso bamboo, *Phyllostachys edulis*, *IDD* family genes, bamboo shoot, alternative splicing

## Abstract

*INDETERMINATE DOMAIN (IDD)* proteins, a family of transcription factors unique to plants, function in multiple developmental processes. Although the *IDD* gene family has been identified in many plants, little is known about it in moso bamboo. In this present study, we identified 32 *PheIDD* family genes in moso bamboo and randomly sequenced the full-length open reading frames (ORFs) of ten *PheIDDs*. All PheIDDs shared a highly conserved IDD domain that contained two canonical C2H2-ZFs, two C2HC-ZFs, and a nuclear localization signal. Collinearity analysis showed that segmental duplication events played an important role in expansion of the *PheIDD* gene family. Synteny analysis indicated that 30 *PheIDD* genes were orthologous to those of rice (*Oryza sativa*). Thirty *PheIDDs* were expressed at low levels, and most *PheIDDs* exhibited characteristic organ-specific expression patterns. Despite their diverse expression patterns in response to exogenous plant hormones, 8 and 22 *PheIDDs* responded rapidly to IAA and 6-BA treatments, respectively. The expression levels of 23 *PheIDDs* were closely related to the outgrowth of aboveground branches and 20 *PheIDDs* were closely related to the awakening of underground dormant buds. In addition, we found that the *PheIDD21* gene generated two products by alternative splicing. Both isoforms interacted with PheDELLA and PheSCL3. Furthermore, both isoforms could bind to the *cis*-elements of three genes (PH02Gene17121, PH02Gene35441, PH02Gene11386). Taken together, our work provides valuable information for studying the molecular breeding mechanism of lateral organ development in moso bamboo.

## 1. Introduction

During the development of multicellular organisms, transcription factors (TFs) are responsible for dictating the fate of individual cells by affecting gene expression. Among them, the *INDETERMINATE DOMAIN (IDD)* family is a class of TFs found only in higher plants. It was first identified through transposon insertion and acts non-cell-autonomously to regulate the production of a transmissible signal in the leaf that elicits the transformation of the shoot apex to reproductive development in maize (*Zea mays*) [1].

The IDDs are highly conserved TFs in both monocots and dicots; they function in multiple developmental processes [2,3,4]. The IDDs are characterized by a N-terminal ID domain that contains four zinc fingers (ZFs) and a long variable sequence for protein interaction [5]. The four ZFs can be further divided into the C2H2-type ZFs (ZF1 and ZF2), which are dedicated to DNA interaction, and the C2HC-type ZFs (ZF3 and ZF4). ZF3 and ZF4, especially ZF4, are indispensable for the interactions between IDDs and the SHORT-ROOT (SHR)-SCARECROW (SCR) complex [6]. Sixteen *IDD*s have been identified in *Arabidopsis*, and twelve have been functionally characterized [3,4]. AtIDD1/ENHYDROUS (ENY) is required for seed maturation and germination [7]. AtIDD3/MAGPIE (MAG) and AtIDD10/JACKDAW (JKD) either physically or genetically interact with the GRAS (GAI, RGA and SCR) domain proteins SHR and SCR to regulate root development or patterning [8,9,10]. AtIDD4/IMPERIAL EAGLE and several other AtIDDs establish cell fate in leaves and other lateral organs [11]. AtIDD14/CUF1, AtIDD15/SHOOT GRAVITROPISM5 (SGR5), and AtIDD16/FALCON cooperatively regulate lateral organ morphogenesis and gravitropism by promoting auxin biosynthesis and transport [12]. *AtIDD14* generates two isoforms by alternative splicing (AS). AtIDD14α accumulates under normal conditions, whereas AtIDD14β accumulates specifically in cold temperatures [4]. Six IDD members, AtIDD2/CARRION CROW/GAF1, AtIDD3, AtIDD4, AtIDD5/RAVEN, AtIDD9/BALDIBIS, and AtIDD10, have been reported to control gibberellin (GA) homeostasis and signaling and modulate flowering time by interacting with DELLAs in *Arabidopsis* [5,13]. AtIDD6/BLUEJAY has been identified as a GA regulator [14], and AtIDD8/NUTCRACKER (NUC) influences photoperiodic flowering by modulating sugar transport and metabolism [15].

IDDs are also essential for plant growth and development in Gramineae species, including rice, maize, and barley (*Hordeum vulgare*). For example, rice OsID1 and maize ZmID1 both play essential roles in flowering; they regulate the transition from vegetative to reproductive growth [16,17,18,19,20]. Ghd10, the rice ortholog of ZmID1, plays an important role in regulating yield component traits by increasing rice plant height and the number of primary branches under short-day (SD) conditions [21]. OsIDD3/ROC1 positively regulates cold stress responses by directly binding to the promoter of the gene that encodes dehydration-responsive element-binding protein (DREB)/CBF1, which is involved in cold tolerance [22]. A duplicated gene pair, *ZmIDDveg9/NAKED ENDOSPERM1* (*NKD1*) and *ZmIDD9*/*NKD2*, regulates aleurone cell specification and differentiation [23]. BROAD LEAF1 (BLF1), which is homologous to AtIDD14, AtIDD15, and AtIDD16, functions during the outgrowth of leaf primordia to limit cell proliferation in the leaf-width direction [24]. In-depth studies have shown that IDD functions by binding to the core *cis*-element TTTGTC in maize, Arabidopsis, and rice [15,25,26,27].

In general, IDD family members have been implicated in many biological functions, including morphogenesis, carbohydrate metabolism, cellular patterning, and hormonal signaling, as well as cell growth, division, and differentiation [3,4,5,7,9,10,12,15,28,29,30]. Although we identified 9 *IDD* genes from the database (V1.0) and performed a simple preliminary analysis of them (in Chinese), many previous RNA sequencing studies have failed to detect *IDD*s in moso bamboo because of their low expression levels [31,32,33,34,35]. Therefore, *IDD* genes have not yet been systematically characterized in moso bamboo (*Phyllostachys edulis*).

Moso bamboo, one of the fastest growing plants on earth, is an important source of income in large areas of Asia and Africa. It is primarily propagated asexually from lateral buds on rhizomes that develop into new shoots, which grow to an average culm height of 13 m. Therefore, the development of lateral buds is an essential step in the entire growth and development process of moso bamboo.

With the rapid development of sequencing technology, the identification and specific molecular functions of IDDs have been validated in the model plants *Arabidopsis* and rice and the C_4_ plant maize. Release of the moso bamboo chromosome-level reference genome provides a great opportunity for understanding this gene family [36].

In the present work, we identified 32 *PheIDD* members in moso bamboo and comprehensively analyzed their phylogenetic relationships, sequence features, interacting proteins, and scaffold distribution. We systematically characterized their expression patterns in underground bamboo buds/shoots of different sizes, aboveground branches of different sizes, and different tissues. We also tested their responses to plant hormone treatments. Finally, we examined the interactions of PheIDD21 isoforms with GRAS proteins as well as their target genes. This comprehensive analysis of *PheIDD* members provides valuable information for further functional characterization.

## 2. Results

### 2.1. Identification and Characterization of PheIDD Family Genes in Moso Bamboo

To identify *IDD* genes in moso bamboo, we performed local blastn and blastp searches using *IDD* sequences from *Arabidopsis* and riceagainst the moso bamboo reference genome (V2.0) [36]. After manually removing sequences with incomplete conserved IDD domains, 32 candidate *PheIDD* genes were obtained (Appendix A). All these 32 PheIDD proteins contained two C2H2 and two C2HC zinc finger (ZF) motifs.

We then amplified the coding sequences of the *PheIDD*s and analyzed their gene structures. The genes were named *PheIDD1* to *PheIDD32* based on their homology to *ZmID1*. Most *PheIDD* genes contained three exons and two introns with normal GT-AG splicing sites (Figure 1). The predicted chemical characteristics of all PheIDDs are summarized in Appendix A. The 32 *PheIDDs* were distributed unevenly on 16 scaffolds, and scaffolds 13 and 14 each contained four *PheIDDs* (Appendix A).

Multiple sequence alignment revealed that all PheIDD proteins contained two C2H2 (ZF1 and ZF2) and two C2HC (ZF3 and ZF4) ZFs, as well as a putative N-terminal nuclear localization signal (NLS, KK/RK/RR) (Figure 2), similar to other IDD proteins from Arabidopsis, rice, and maize. Their amino acid sequences displayed an overall identity of 32.83%, and the highly conserved IDD domain exhibited an identity of 75.41%. We also identified two C-terminal domains: the TR/L/QDFLG domain that is characteristic of IDDs from higher plants and an MSATALLQKAA domain that is present in some IDDs but absent from others (Appendix A). In addition, a spacer region longer than those between ZF2 and ZF3 and between ZF3 and ZF4 was identified between ZF1 and ZF2: 37 aa in PheIDD20, 34 aa in PheIDD23, and 29 aa in PheIDD1 and PheIDD2 (Figure 2).

Phosphorylation site analysis identified two conserved amino acid residues, Ser73 and Ser182 among all the PheIDDs (Appendix A). This result suggests that these two amino acids may be necessary for PheIDD function.

### 2.2. Phylogenetic Analyses, Synteny, and Duplication of the PheIDDFamily Genes

A phylogenetic tree of the 32 PheIDD proteins was constructed using the NJ method, and nine groups were identified (Figure 1). Group 1 contained eight *PheIDDs* that formed four putative paralogous gene pairs (*PheIDD13*/*PheIDD18*, *PheIDD15*/*PheIDD17*, *PheIDD19*/*PheIDD26*, and *PheIDD21*/*PheIDD25*). Group 2 had six PheIDD proteins that formed three putative paralogous gene pairs (*PheIDD27*/*PheIDD29*, *PheIDD28*/*PheIDD30*, and *PheIDD20*/*PheIDD23*). Group 3 had five PheIDD proteins, and two putative paralogous gene pairs (*PheIDD5*/*PheIDD6* and *PheIDD9*/*PheIDD22*) were identified. Group 4 had four PheIDD proteins that formed two putative paralogous gene pairs (*PheIDD31*/*PheIDD32* and *PheIDD10*/*PheIDD14*). Groups 5–7 each contained only one putative paralogous gene pair (*PheIDD11*/*PheIDD12*, *PheIDD1*/*PheIDD2*, and *PheIDD7*/*PheIDD16*, respectively). PheIDD4 and PheIDD24 belonged to group 8. *PheIDD8* was clustered into a separate group by itself. Fourteen putative paralogous gene pairs were determined based on Ka/Ks ratios, and 29 orthologous pairs of *IDDs* between rice and moso bamboowere also defined using this method (Table 1).

To reveal the evolutionary relationships between *IDD* genes from bamboo and other plant species, an unrooted NJ phylogenetic tree was constructed by aligning full-length IDD proteins from moso bamboo, rice, maize, and Arabidopsis. The IDD proteins were resolved into 15 distinct groups, each containing an OsIDD (Figure 3). In most cases, the orthologous groups were monophyletic, and we therefore could not determine one-to-one orthologous relationships between Arabidopsis and moso bamboo IDD family members. By contrast, one-to-two rice/moso bamboo *IDD* gene pairs could be identified. A comparative synteny analysis was performed to explore the origin and evolution of the *PheIDD* genes. Orthologous relationships were detected between 30 *PheIDD* genes and 15 *OsIDD* genes, most of which were located in syntenic loci on the rice chromosomes and moso bamboo scaffolds (Figure 4). Two *PheIDD* genes were unique to moso bamboo. Consistent with the phylogenetic tree, multiple *PheIDD* genes were identified as putative orthologs of a single *OsIDD* gene. These results indicated that expansion of *PheIDD* genes may have occurred after the divergence of the lineages leading to rice and moso bamboo.

In addition, tandem and segmental duplication events were investigated to provide additional insight into *PheIDD* gene duplication. No *PheIDD* genes were identified as tandem duplicates, whereas 22 segmental duplication events were identified (Appendix A). These results indicated that expansion of the *PheIDD* genes may have occurred primarily through segmental duplication.

### 2.3. Expression Profiles of PheIDDs in Different Tissues

The *IDD* genes have been reported to display obvious tissue-specific expression patterns. For example, some *IDDs* are preferentially expressed in mature leaves [37], some in immature leaves [16,20] or roots [15]. To determine the tissue-specific expression patterns of the *PheIDDs*, roots, stems, mature leaves, immature leaves, and tillers of 2-month-old moso bamboo seedlings were used for qRT–PCR analysis. With the exception of *PheIDD21* and *PheIDD25*, other *PheIDDs* exhibited very low transcript levels in all tested tissues (Figure 5). Most *PheIDDs* showed an obvious organ-specific expression pattern. For example, *PheIDD21* was exclusively expressed in tillers, whereas *PheIDD25* accumulated preferentially in roots. Only two genes (*PheIDD2* and *PheIDD21*) had their highest expression level in tillers. Ten *PheIDDs* had relatively higher transcript levels in leaves, including mature and immatureleaves: five (*PheIDD3*, *PheIDD13*, *PheIDD15*, *PheIDD17*, and *PheIDD26*) were preferentially expressed in mature leaves and four (*PheIDD19*, *PheIDD22, PheIDD31*, and *PheIDD32*) in immature leaves. *PheIDD6* had similar transcript levels in mature and immature leaves. Seventeen genes had relatively higher expression levels in roots than in other tissues, and seven of these (*PheIDD1*, *PheIDD4*, *PheIDD7*, *PheIDD12*, *PheIDD24*, *PheIDD25*,and *PheIDD30*) were preferentially expressed in roots. Collectively, these data suggest that most *PheIDDs* display strong tissue-specific expression patterns, and five pairs of paralogous genes, such as *PheIDD7*/*PheIDD16* that are highly expressed in roots, share similar expression patterns.

### 2.4. Dynamic Expression Profiles of PheIDDs in Underground Lateral Buds/Shoots at Different Developmental Stages

In *Arabidopsis*, IDD family members perform diverse functions, including lateral shoot morphogenesis and cellular patterning [11,12]. To further characterize their functions during the development of underground lateral buds/shoots in bamboo (Figure 6a), we analyzed the expression patterns of all *PheIDDs* by qRT–PCR. Twenty of the 32 *PheIDD* genes were upregulated in underground buds upon dormancy awakening (from stage S1 to S2), and the expression of 14 genes reached the highest level at the S2 stage. The expression of *PheIDD1* showed the greatest increase (39.4-fold) (Figure 6b and Appendix A). By contrast, transcript levels of the *PheIDD11*/*PheIDD12* gene pair were reduced from dormant buds (S1) to 16-cm buds (S4), suggesting that they may be related to the maintenance of bud dormancy. In addition, a paralogous gene pair (*PheIDD21*/*PheIDD25*) displayed opposite expression patterns in underground bamboo buds/shoots of different sizes.

### 2.5. Dynamic Expression Profiles of PheIDDs in Aboveground Branches of Different Sizes

To further characterize their functions during the development of branches in moso bamboo, expression patterns of all *PheIDDs* were analyzed in aboveground branches of different sizes (Figure 7a) by qRT–PCR. Twenty-three *PheIDDs* were expressed at a very low level in the dormant bud; their transcript levels peaked in 1-cm branches (L2) and then declined when branch length reached 7 cm (L4) (Figure 7b and Appendix A). The absolute transcript levels of 23 *PheIDDs* increased more than two-fold in 1-cm branches compared with dormant buds, and *PheIDD1* showed the greatest increase (39.3-fold). By contrast, the expression levels of five *PheIDDs* (*PheIDD5*, *PheIDD25*, *PheIDD26*, *PheIDD31*, and *PheIDD32*) were very high in dormant buds and then gradually decreased with bud development. A total of 28 *PheIDDs* were expressed at high levels in aboveground branches; these included *PheIDD2* and *PheIDD21*, which had higher expression levels in tillers than in other tissues. Several paralogous gene pairs, including *PheIDD1*/*PheIDD2*, showed a similar expression pattern, whereas others exhibited different or even opposite expression patterns. These observations reflect the different evolutionary fates of duplicated genes.

### 2.6. Transcript Profiling of PheIDDs under IAA and 6-BA Treatments

Previous studies have reported a link between IDD members and hormones. IDD members were documented to mediate hormone signaling to regulate plant development [5,12,38,39]. A *cis*-element analysis showed that each *PheIDD* promoter contained at least one *cis*-element implicated in the response to hormones. To investigate whether *PheIDDs* are involved in hormone signaling, we analyzed the expression patterns of each *PheIDD* under hormone treatments by qRT–PCR.

As shown in Figure 8, 21 *PheIDD*s showed strong responses to exogenous 6-BA. The expression levels of eight *PheIDDs* (*PheIDD10*, *PheIDD12*, *PheIDD13*, *PheIDD20*, *PheIDD23*, *PheIDD25*, *PheIDD27*, and *PheIDD28*) increased significantly, whereas those of thirteen *PheIDDs* (*PheIDD1*, *PheIDD2*, *PheIDD3*, *PheIDD6*, *PheIDD16*, *PheIDD17*, *PheIDD18*, *PheIDD19*, *PheIDD22*, *PheIDD26*, *PheIDD30*, *PheIDD31*, and *PheIDD32*) were lower at four or more time points in treated compared with untreated seedlings. *PheIDD25* expression was either unchanged or higher in the 6-BA treatment than in the control at all time points, whereas *PheIDD22* displayed the opposite expression pattern in the 6-BA treatment. The expression levels of 16 *PheIDDs* were highest 2 h after 6-BA treatment, indicating that they respond rapidly to 6-BA.

Fourteen *PheIDDs* responded strongly to exogenous IAA application (Figure 9). Specifically, the expression levels of nine *PheIDDs* (*PheIDD3*, *PheIDD9*, *PheIDD10*, *PheIDD12*, *PheIDD13*, *PheIDD20*, *PheIDD24*, *PheIDD26*, and *PheIDD30*) increased significantly, whereas those of five *PheIDDs* (*PheIDD2*, *PheIDD6*, *PheIDD16*, *PheIDD17*, and *PheIDD31*) decreased in IAA-treated samples compared with the control at four or more time points. The transcript levels of *PheIDD25*, *PheIDD28*, and *PheIDD30* were either unchanged or increased upon IAA treatment compared with the control at all eight time points. One paralogous gene pair, *PheIDD5*/*PheIDD6*, displayed similar expression patterns inIAA-treated samples. Taken together, these results revealed different responses of many duplicated *PheIDD* gene pairs to hormone treatments.

### 2.7. PheIDD21Generated Two Transcripts by Alternative Splicing

During cloning of the *PheIDD* genes, we unexpectedly found that two PCR products were generated by the *PheIDD21* primer pair, suggesting AS of *PheIDD21* in moso bamboo. By comparing the two cDNA sequences, we discovered that one splicing product was missing 9 nucleotides at the 3′ end of the first exon compared with the full-length transcript (Figure 10a). The AS type was therefore an alternative donor site (selection of 5′ splice sites). We named the full-length transcript PheIDD21α and the shorter transcript PheIDD21β. A protein sequence alignment showed that the three missing amino acids were located next to the NLS but not within the conserved domain.

### 2.8. Interaction of PheIDD21 with RGAS Proteins and Its Binding to Target Genes

IDD proteins have been reported to form homo- and heterodimers with proteins from different families, especially the GRAS TFs [4]. To identify GRAS proteins that interact with IDDs, we searched a protein–protein network using STRING 10 software. Given that none of the 16 Arabidopsis IDDs were homologous to ZmID1/OsID [16], we used the full-length sequences of 30 PheIDD proteins (PheIDD3–PheIDD32) to search for candidate interacting proteins based on their interologs in the Arabidopsisinteractome (Appendix A). A total of 11 high-confidence interacting proteins were identified for the IDDs, four of which, SCARECROW-LIKE3 (SCL3), SCR, SHR, and REPRESSOR of *ga1-3* (RGA), belonged to the GRAS family. All 30 PheIDD proteins, including PheIDD21, were predicted to interact with AT2G20050, a PP2C phosphatase. In addition, five PheIDDs, including PheIDD21, were shown to interact with SCL3 and RGA1/DELLA. We then performed yeast two-hybrid (Y2H) experiments to test the predicted interactions. Yeast cells were co-transformed with PheIDD21 isoforms and GRAS proteins, including PheSCL3 and PheDELLA. All yeast cells grew normally on SD/-Trp-Leu medium.

The positive control and various tested protein combinations grew normally on SD/-Trp-Leu-His-Ade medium, whereas the negative control did not (Figure 10b). The results showed that both PheIDD21α and PheIDD21β could interact strongly to form homodimers and could also form heterodimers. Both isoforms could form heterodimers with PheDELLAa, PheDELLAb, and PheSCL3a. The data therefore confirmed that both PheIDD21 isoforms could function as homodimers and heterodimers.

IDD members act as TFs to regulate a variety of plant developmental and physiological processes [3], and we therefore investigated their downstream target genes. First, we constructed a gene co-expression network with PheIDD21 based on the transcriptome data (Appendix A). Previous studies have shown that IDD TFs directly bind to the TTTGTC core element [15,25,26], and we next analyzed *cis*-elements in the genome sequences of the co-expressed genes. Four putative elements in the promoter and intron of gene PH02Gene17121, three putative elements in the promoter and intron of gene PH02Gene35441, and two putative elements in the promoterof gene PH02Gene11386 were identified (Figure 11a). We next performed yeast one-hybrid (Y1H) assays with 220–350-bp fragments to test the binding affinity of PheIDD21 isoforms to the above three genes. Both PheIDD21α and PheIDD21β bind to three (A, B, and D) of the four elements of gene PH02Gene17121, one (A) of the three elements of gene PH02Gene35441, and one (A) of the two elements of gene PH02Gene11386 (Figure 11b), suggesting that PheIDD21 directly binds to these three genes to regulate their expression.

## 3. Discussion

The IDD family is a plant-specific zinc finger subfamily. Previous reports haveshown that several IDD members play essential roles in the development of Arabidopsis, rice, and maize; however, little is known about IDD members in moso bamboo. In the current study, we performed a genome-wide investigation of the *IDD* gene family in moso bamboo.

### 3.1. The IDDGene Family in Moso Bamboo

Prochetto and Reinheimer reported that the IDD TF originated in the last common ancestor of Charophytic algae and land plants and underwent a gene duplication event ~470 million years ago (MYA) [2]. The *IDDs* were then duplicated several times during the emergence and diversification of land plants. Therefore, the monocot and dicot genomes contain similar numbers of *IDD* genes (16–23) [16,40,41]. However, monocots have a higher number of IDD clusters than dicots, most of which are specific to grass families; one example is the OsID1/RID1/Ehd2 cluster [2]. This suggests that whole-genome duplication (WGD) may have given rise to the grass *IDD* gene families [42,43]. Here, we identified 32 *PheIDD* genes in moso bamboo (Figure 1 and Figure 3), suggesting that additional duplications, especially segmental duplication events, have occurred in the *IDD* family during bamboo evolution (Appendix A). Gene duplication and diversification are important for plant evolution and have been reported to contribute to the morphological evolution of moso bamboo [44]. Although the NLS motif was reported to be absent from some IDD proteins [2], all PheIDDs contained this motif (Figure 2). Therefore, it is very likely that PheIDDs are located in the nucleus and function astranscriptional regulators. Furthermore, all PheIDDs are likely to be functional because, like Arabidopsisand maize IDD proteins, they each possess all four intact ZFs (two C2H2 ZFs and two C2HC ZFs) [1,37]. In addition, they all contained two conserved amino acid residues (Appendix A): Ser73 has been reported to be modified by MPK6 in several independent phosphoproteomic studies [45], and Ser182 is known to be phosphorylated by AKIN10 [38]. Although the PheIDDs share high similarity in their N-terminal ID domains (75.41% identity), the remaining portions of the sequences vary substantially. An exception occurs in two conserved regions at the C terminus of 21 PheIDDs (Appendix A), which are exclusively found in the IDDsof higher plants [16].

Both the IDD phylogenetic tree and the full-length sequence alignment suggest that PheIDD1 is closely related to ZmID1 (64.57% identity) and OsID1 (61.59% identity) [16]. Like ZmID1 and OsID1, PheIDD1 also contains a long spacer region of 29 aa between the first two ZFs (Figure 2) [16]. However, the tissue-specific expression pattern of *PheIDD1* differed from those of *OsID1* and *ZmID1*, indicating thatit may function differently. Whether this feature of *PheIDD1* is related to the irregular flowering of moso bamboo is worthy of in-depth exploration.

Three of the 16 AtIDDs (AtIDD14, AtIDD15, and AtIDD16) and three of the 15 OsIDDs (OsIDD12, OsIDD13, and OsIDD14/Loose Plant Architecture1, LPA1) shared a unique conserved coiled-coil domain at the Cterminus but had lost two other conserved domains (MSATALLQKAA and TR/LDFLG), at least one of which is present in other IDD members [16]. The six PheIDDs that are homologous to the above proteins (PheIDD20, PheIDD23, PheIDD27, PheIDD28, PheIDD29, and PheIDD30)—as well as two other PheIDDs (PheIDD10 and PheIDD14)—have all lost these two conserved C-terminal domains, suggesting that these 14 proteins may have similar functions (Appendix A).

### 3.2. PheIDDs Respond to Exogenous Hormone Treatments

Previous studies have shown that some IDDs are involved in hormone signaling as well as hormone homeostasis to regulate plant development [5,38,45]. AtIDD14, AtIDD15, and AtIDD16 are known to cooperatively regulate lateral organ morphogenesis and gravitropism by promoting auxin biosynthesis and transport [12]. Conversely, some *IDD*s are subject to regulation by hormones [22,46]. Gamuyao et al. have reported that in addition to auxin, CTK and GA also accumulate in the moso bamboo shoot apical meristem (SAM), and several *PheIDD* genes are highly expressed in the SAM, suggesting a possible link between *IDDs* and these hormones [47]. Our data showed that the expression of 24 of the 32 *PheIDDs* was induced by 6-BA, and 22 reached the maximum fold-change in less than 6 h (Figure 8). The expression of 19 of the 32 *PheIDDs* was induced by IAA, and the expression of eight *PheIDDs* peaked less than 6 h after hormone application (Figure 9). Collectively, our data indicate that *PheIDDs* respond rapidly to hormone treatments and that they may play roles in the development of bamboo shoots.

### 3.3. Expression Profile Analysis of PheIDDGenes in the Development of Aboveground Branches, and Underground Buds/Shoots

All *IDDs* have undergone extensive gene duplications in monocots during evolution and have acquired diverse expression patterns [2]. Most *IDDs*, including duplicated *IDDs*, show different expression levels during different stages of vegetative organ development [2]. Here, *PheIDD25* exhibited the highest expression level in all tissues tested, especially in roots, whereas its paralog, *PheIDD21*, had a moderate expression level in all tissues tested. The remaining 30 *PheIDD* genes were expressed at low levels in the tested tissues (Figure 5). In addition, more than half of the *PheIDDs* had a high expression level in roots, and ten and three *PheIDDs* accumulated primarily in leaves and stems, respectively. The remaining two genes, *PheIDD2* and *PheIDD21*, were highly expressed in tillers, whereas their paralogs *PheIDD1* and *PheIDD25* were expressed exclusively in roots. These data suggest that the functions of each pair of parologous genes may have diverged during the evolution of grass species.

AtIDD4 and several other IDDs have been implicated in the establishment of adaxial/abaxial patterns and cell fates in leaves and other lateral organs [11]. Mutation of *IDD4* and *IDD4* overexpression have been reported to affect the shoot and root growth of Arabidopsis [45]. AtIDD14, AtIDD15, and AtIDD16are known toplay critical roles in lateral shoot morphogenesis and gravitropism by regulating spatial auxin accumulation [12]. Therefore, we assume that some PheIDDs may be related to the development of bamboo buds/shoots in moso bamboo. Consistent with this hypothesis, the expression levels of 20 *PheIDDs* (62.5%) were higher in awakening buds compared with dormant buds (Figure 6). The above data suggest that several PheIDD TFs may be involved in the early development of underground bamboo buds/shoots. Given the changes in *PheIDD* expression during the transition from underground dormant buds to awakened buds, as well as the conserved C-terminal motifs shared by *PheIDD25* and *PheIDD26*, we speculate that these two proteins may function redundantly in the development of underground bamboo buds/shoots. This possibility remains to be investigated in future studies. Consistent with the results obtained by Gamuyao et al. [47], *PheIDD18* expression increased from S1 to S4 and was highest in S4, suggesting a possible role in the later growth and development of underground bamboo shoots (Figure 6b). Expression of the *PheIDD27* gene was similar. However, we did not identify *PheIDDs* whose expression increased in the L4 stage, suggesting that PheIDDs were not involved in the late stages of aboveground branch development.

It is worth noting that although *PheIDD1* is most closely related to rice *OsID1* and maize *ZmID1*, it was primarily expressed in roots and showed the greatest increase in expression compared to other *PheIDDs* during the development from dormant buds to awakening buds. Similar to previous findings for maize *ZmID1*, Wu et al. reported that *OsID1* is exclusively expressed in immature leaves but not in roots, mature leaves, the SAM, or other tissues [20]. Given that OsID1 functions as a master switch that controls the transition from vegetative growth to floral development, our results suggest that the regulation of PheIDD1 may be different from that of OsID1 and that PheIDD1 may be involved in the regulation of lateral shoot development.

### 3.4. IDD Interacting Proteins and Target Genes

Based on phylogenetic analysis, *PheIDD21* has the highest homology with rice *OsIDD10* and with a maize duplicated gene pair, *ZmIDDveg9* and *ZmIDD9*. *OsIDD10* is expressed in the primordia of lateral roots [26], and ZmIDDveg9 and ZmIDD9 are required for cell patterning and differentiation, cell growth and division, and hormone pathways [23,39]. In addition, several Arabidopsis IDDs, including AtIDD10 and AtIDD3, are required for cell division in ground tissues through direct protein–protein interactions with the GRAS proteins SHR, SCR, and SCL3 [9,48,49]. Both isoforms of AtIDD14 interact with each other and participate in starch metabolism [4]. Our work suggests that *PheIDD21* generates two transcripts by AS. Both the isoforms interact with each other to form homodimers and interact with GRAS proteins to form heterodimers (Figure 10).

Multiple studies have reported that IDD TFs bind to *cis*-elements to regulate plant growth and development. AtIDD8 contributes to photoperiodic flowering by binding directly to the CTTTTGTCC sequence motif within the *SUS4* gene promoter [15]. OsIDD10 influences root ammonium uptake and nitrogen metabolism by activating *AMT1;2* and *GDH2*, binding to a specific *cis*-element TTTGTC in the *AMT1;2* promoter and toa similar TTTGTCC/G element in the fifth intron of *GDH2* [26]. OsIDD13 and OsIDD3 regulate defense to sheath blight disease by binding to the TTTGTCG *cis*-element in the *PIN1a* promoter [27]. The maize ID1 protein also binds to the TTTTGTCG/C sequence [25]. Similarly, PheIDD21 in this study bindsto the core *cis*-elements of three genes (Figure 11), suggesting that PheIDD21 may play a role in regulating their expression.

## 4. Methods and Materials

### 4.1. Plant Materials and Treatments

The moso bamboo plantation was located in Qingshan, Lin’an District, Hangzhou City, Zhejiang Province, China (30°14’ N, 119°42’ E). Underground buds or shoots at four different developmental stages (S1–S4) in this area were sampled as described in a previous study from our lab [50]. Buds at the S1 stage were dormant, ~0.2 cm in length, and close to the groove of the rhizome. In stage S2, the angle between the tip of the awakening bud (2 cm in length) and the rhizome was ~45°. In stage S3, shoots were 8 cm in length and more robust, and the angle between the tip of the shoot and the rhizome was 60°. Shoots at the S4 stage were ~16 cm in length. At least three buds or shoots with similar size and shape were harvested from the rhizome at each stage. Bamboo sheaths were carefully removed, and all buds or shoots were collected separately.

Developing branches were harvested from one-year-old moso bamboo plants which had just ended their height growth. Branches at three different developmental stages (1, 4, and 7 cm in length) on the same plant were harvested. The base of an internode groove that was beneath an internode with a branch was collected as the dormant bud (referred to as 0-cm bud in this study). At least three branches of similar size were harvested in each stage.

The seeds of moso bamboo used in this study were collected from Guilin, Guangxi. The seeds were rinsed and soaked in water at room temperature for 24 h and then germinated in Petri dishes on moist, sterile filter paper at room temperature. After growing for 10 days, some seedlings were moved to Hoagland’s nutrient solution in containers. The nutrient solution was changed every two days to ensure seedling growth.

Tissue-specific expression of the *PheIDD*s: Various vegetative tissues, including roots, stems, mature leaves, immature leaves, and tillers, were collected from 2-month-old seedlings. Three biological replicates were analyzed, with five samples per biological replicate.

Expression patterns of the *PheIDD*s under exogenous plant hormones: Moso bamboo seedlings were sprayed with 500 μM IAA or 500 μM 6-BA, and control seedlings were sprayed with water. A 0.6-cm section of the stem base, where tiller buds emerged, was harvested from treated and control plants at 0, 2, 4, 6, 12, 24, 48, and 72 h after plant hormone treatments. Three biological replicates were analyzed, with five sections per replicate.

All samples were immediately frozen in liquid nitrogen and stored at −80 °C until further use.

### 4.2. Identification of PheIDDFamilyGenes in the Moso Bamboo Genome

The nucleotide and protein sequences of 16 *AtIDD*, 15 *OsIDD*, and 23 *ZmIDD* geneswere downloaded from the Phytozome v12.1 database (https://phytozome.jgi.doe.gov/pz/portal.html, accessed on 30 May 2019). These were used as query sequences to search against moso bamboo databases downloaded from GigaDB [36] (https://trace.ncbi.nlm.nih.gov/Traces/sra/sra.cgi, accessed on 31 May 2019) and BambooGDB [31,51] (http://server.ncgr.ac.cn/bamboo/down.php and http://www.bamboogdb.org/, accessed on 31 May 2019) with cutoffs of “e-value ≤ 0.001” and “alignment-length ≥ 100” using Blast (version 2.2.25). Redundant sequences and sequences that lacked the complete IDD domain were removed after a similarity comparison. The resulting sequences were used to search the National Center for Biotechnology Information (NCBI) database to confirm their identities as *IDD*s. Several candidate *PheIDDs* were randomly selected and amplified from the cDNA of moso bamboo, cloned into the pMD18 vector (Takara, Japan), and Sanger-sequenced in both directions to verify their sequences. Gene-specific primers were designed using the online tool Primer3 (http://bioinfo.ut.ee/primer3-0.4.0/, accessed on 16 June 2019) and are listed in Appendix A. Putative *PheIDD* genes were named *PheIDD1* to *PheIDD32* based on their homologyto the maize *ZmID1*.

### 4.3. Bioinformatic Analyses

#### 4.3.1. Analysis of Chemical Characteristics

The chemical characteristics of all predicted proteins, including CDS length, amino acid number, and physicochemical parameters, were analyzed using the online analysis tool ProtParam (http://web.expasy.org/protparam, accessed on 18 August 2019).

#### 4.3.2. Gene structures and Scaffold Locations

Structures of the *PheIDD*s were generated using the redraw gene structure function in TBtools (V1.068), and a scaffold distribution map was generated using the amazing gene location function in TBtools [52].

#### 4.3.3. Screening of Gene Pairs and Calculation of Ka/Ks Ratios

*PheIDD* paralogous in moso bamboo and orthologous *IDD* gene pairs between moso bamboo and rice were identified as described in a previous study [53]. Ka/Ks ratios were calculated using the Simple Ka/Ks Calculator function of TBtools.

#### 4.3.4. Synteny Analysis and Gene Duplication

The moso bamboo protein sequences were aligned to the protein sequences from rice using TBtools software. Duplication events of the *PheIDD* genes were confirmed as described previously [54]. The figures were visualized using Circos and Dual Synteny Plot in TBtools [52].

#### 4.3.5. Prediction of Phosphorylation Sites in the PheIDDs

Phosphorylation sites in the 32 PheIDD proteins were predicted using online NetPhos software (http://www.cbs.dtu.dk/Services/NetPhos/, accessed on 18 August 2019).

#### 4.3.6. Prediction of Protein–Protein Interactions

Protein–protein interactions were predicted using the STRING online platform (http://string-db.org/, accessed on 18 March 2020) based on interologs from the *Arabidopsis* database.

#### 4.3.7. Multiple Sequence Alignment and Phylogenetic Analyses

Multiple sequence alignment of the 32 PheIDD proteins and the maize ZmID1 protein was performed using DNAMAN software (Version 6). Sequences of the rice, maize, and *Arabidopsis* IDD proteins were downloaded from the Phytozome v12.1 database (https://phytozome.jgi.doe.gov/pz/portal.html, accessed on 18 November 2019).The accession numbers of the 86 *IDDs* used in this study are provided in Appendix A. The amino acid sequences of IDD proteins were aligned using Clustal W software. A phylogenetic tree was constructed using the NJ method implemented in MEGA6 with 1000 bootstrap replicates.

### 4.4. Quantitative Real-Time PCR (qRT–PCR)

Total RNA was extracted using the TRIzol reagent (Invitrogen, Waltham, MA, USA), and RNase-free DNase I (Promega, Madison, WI, USA) was used to remove genomic DNA following the manufacturer’s instructions. cDNA was synthesized with 2 μg of RNA using a PrimeScript RT kit (Takara, Shiga, Japan) following the manufacturer’s recommendations. Gene-specific primers for the 32 *PheIDD* genes were designed using the online Primer3 tool. Due to the high similarity of PheIDD N-terminal sequences, we avoided these conserved regions when designing primers. Primers were designed in the 5′ and 3′ UTR regions to ensure specificity when necessary. PCR products obtained using each primer pair were validated by Sanger sequencing.

The qRT–PCR experiment was performed using the SYBR Color qPCR Master Mix reagent (Vazyme, Nanjing, China) and the CFX 96 Real-Time System (Applied Biosystems, Waltham, CA, USA). The 15-µL qRT–PCR assay mix contained 1.2 µL cDNA (diluted 1:8), 7.5 µL 2 × SYBR Color qPCR Master Mix reagent (Vazyme), 0.5 µL of each primer (100 μM), and 5.3 µL distilled deionized H_2_O. The reaction procedure was as follows: initial denaturation at 95 °C for 3 min, followed by 39 cycles of 95 °C for 10 s and 58 °C for 30 s. The specificity of the amplified products was verified by a melting-curve analysis. The *PheNTB* gene (GenBank accession No. gi|242381788) was used as the internal control to normalize the transcript levels of target genes [55]. The relative transcript levels of target genes were calculated and analyzed using the 2^−ΔΔCt^ method [56]. Three biological replicates and three technical replicates were analyzed for each sample.The primers are listed in Appendix A.

### 4.5. Yeast Two-Hybrid Assay (Y2H)

The CDSs of *PheIDD21α* and *PheIDD21β* were cloned into the pGBKT7 vector, and the CDSs of three predicted target genes (*PheDELLAa*, *PheDELLAb*, and *PheSCL3*) were cloned into the pGADT7 vector. The pGBKT7 vector and pGADT7 vector were co-transformed into the yeast strain AH109. The co-transformed yeast was cultured in SD/-Trp/-Leu (-DDO) agar medium at 30 °C for 3 days and then transferred to SD-Trp-Leu-His-Ade (-QDO) for 3–5 days at 30 °C to identify the interaction in yeast. Y2H assays were performed according to the Clontech Matchmaker GAL4 Two-Hybrid System 3 & Libraries User Manual instructions. The positive and negative controls were from the kits. The primers are listed in Appendix A.

### 4.6. Yeast One-Hybrid Assay (Y1H)

The CDSs of *PheIDD21α* and *PheIDD21β* were cloned into the prey vector (pGADT7), and the 230–340-bp promoter fragments of IDD-binding sites (four putative elements in PH02Gene17121, three putative elements in PH02Gene35441,and two putative elements in PH02Gene11386) were amplified and ligated into the pHIS2 vector (Clontech). The corresponding pGADT7 and pHIS2 constructs were co-transformed into yeast strain Y187 following the manufacturer’s instructions for the yeast one-hybrid library screening system (Clontech). Interaction of pHIS53 with pGADT753 was used as a positive control, and the empty AD and pHIS-*pro*-frag were used as negative controls. The transformed strains were cultured at 30 °C for 2–3 days on SD/-Trp/-Leu media and were used for spot assays on SD/-Trp/-Leu/-His media with 10 mM 3-AT.The primers are listed in Appendix A.

### 4.7. Statistical Analysis

The statistical analysis of all data in this study was carried out in SPSS 11.5 software (Chicago, IL, USA). Analysis of variance (the GLM procedure) was used to analyze the data. Data are presented as means ± standard deviation (SD). Graphs were generated using Sigmaplot 12.5 (Systat Software Inc., San Jose, CA, USA).

## 5. Conclusions

In this study, we identified and characterized 32 PheIDD proteins that contained a complete IDD domain. They were unevenly distributed among 16 scaffolds. Based on their amino acid sequences, the 32 PheIDDs were divided into nine groups. Synteny analysis indicated that 30 *PheIDD* genes were orthologous to those in rice. During the evolution of *PheIDD* genes, segmental duplication events played an important role. Expression analysis revealed the diverse expression patterns of *PheIDDs* in different tissues. In addition, *PheIDDs* responded differently to exogenous applications of the plant hormones IAA and 6-BA, and their expression patterns were different during the development of aboveground branches and underground buds/shoots. A few *PheIDD* genes were predicted as candidate genes to study their functions during bamboo shoot development. Finally, we determined that one candidate gene, *PheIDD21*, underwent AS and generated two transcripts. Both isoforms formed homodimers and heterodimers and bound to the *cis*-elements of three genes. These findings increase our understanding of the characteristics of the *PheIDD* gene family and provide valuable information for further functional characterization of *PheIDDs* in the shoot development of moso bamboo.

## Figures and Tables

**Figure 1 ijms-23-13952-f001:**
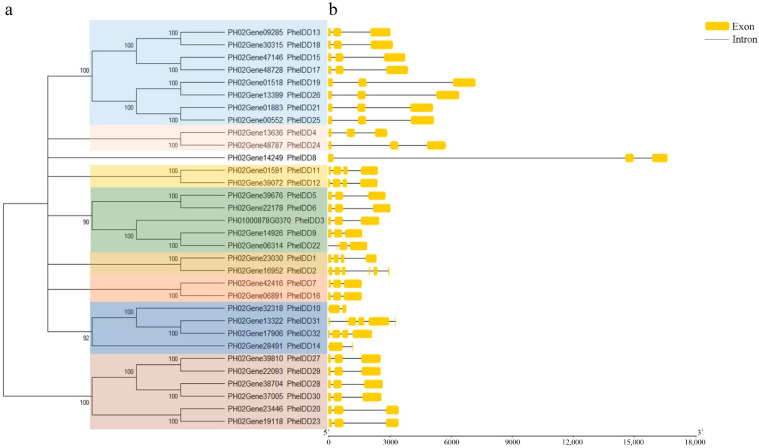
Phylogenetic analysis and schematic gene structures of *IDDs* in moso bamboo. (**a**) A phylogenetic tree of IDD homologs in moso bamboo. The tree was constructed by the neighbor-joining method using MEGA6.0 software with default parameters. Numbers at the nodes indicate bootstrap values (as percentages) calculated for 1000 replicates. Sub-groups are highlighted with colored boxes.(**b**) Schematic representation of *PheIDD* genes. Exons are shown as boxes and introns as lines.

**Figure 2 ijms-23-13952-f002:**
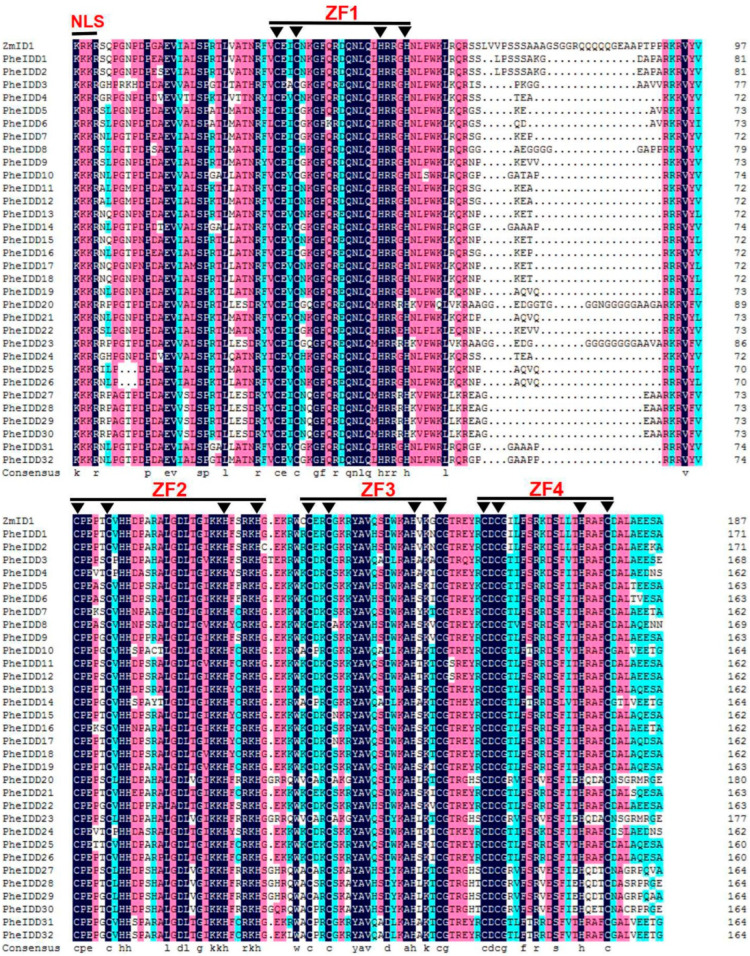
Alignment of amino acid sequences of 32 PheIDDs and ZmID1 proteins. The four zinc-finger motifs are marked with straight lines. Inverted triangles indicate conserved amino acids, CCHH in ZF1 and ZF2, and CCHC in ZF3 and ZF4. NLS at the N-terminal region shows the putative NLS motif. Identical amino acids are shown with text on a black background.

**Figure 3 ijms-23-13952-f003:**
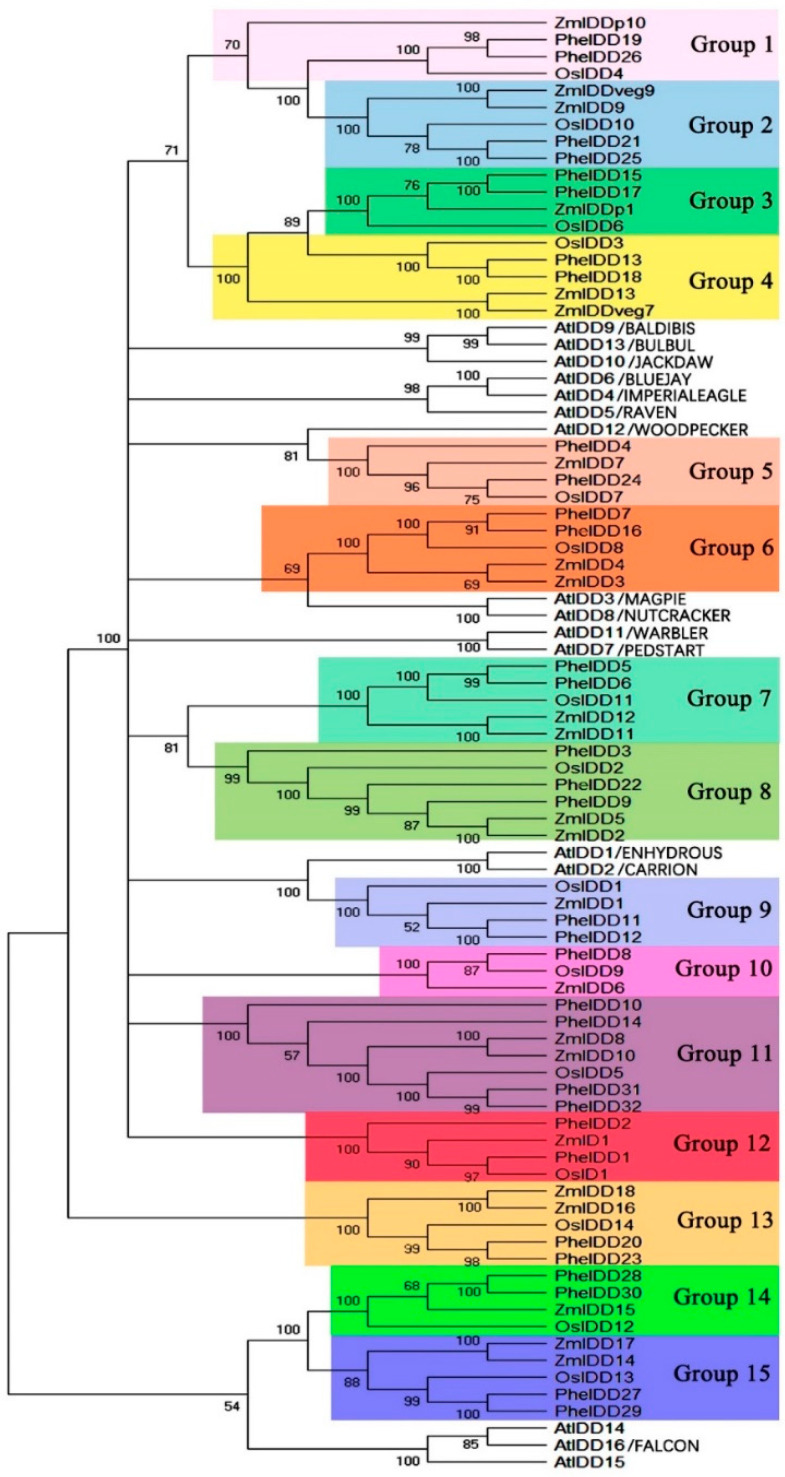
Phylogenetic analysis of IDD homologs from four different plant species: *Arabidopsis thaliana*, *Z. mays*, *O. sativa*, and *Ph. edulis*. The tree was constructed by the neighbor-joining method using MEGA6.0 software with default parameters. Bootstrapping was performed 1000 times, and percentage support values are shown on branches. Putative orthologous groups are highlighted with colored boxes. The accession numbers of the genes used for sequence analysis and tree construction are shown in Appendix A.

**Figure 4 ijms-23-13952-f004:**
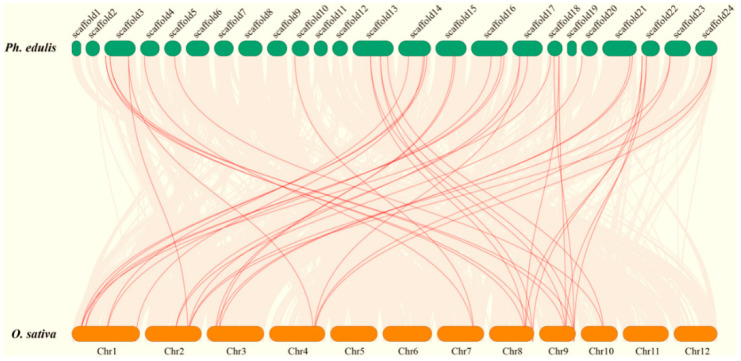
Synteny analysis of the moso bamboo and rice genomes. The gray lines represent aligned blocks between the paired genomes, and the red lines indicate syntenic *IDD* gene pairs.

**Figure 5 ijms-23-13952-f005:**
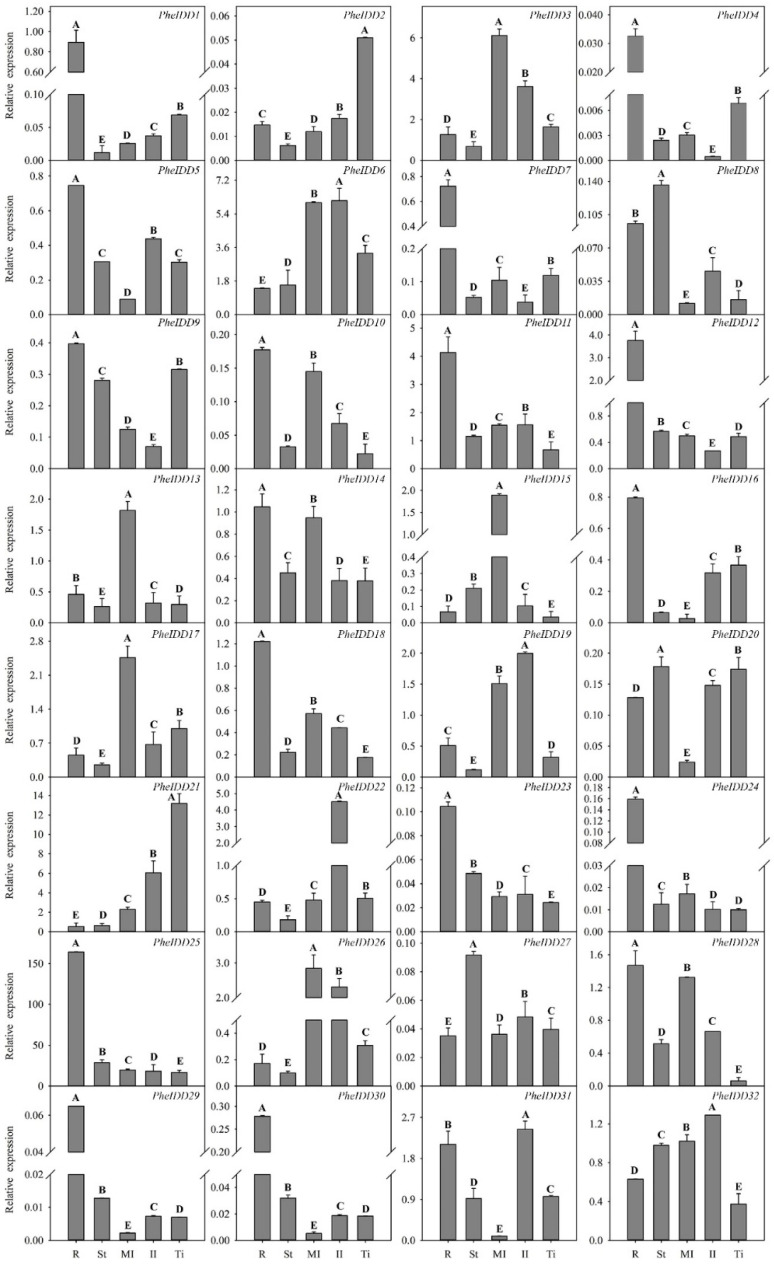
Tissue-specific expression patterns of *PheIDD* family genes in different tissues of moso bamboo. R: root, St: stem, Ml: mature leaf, Il: immature leaf, Ti: tiller. The *PheNTB* gene was used as a reference. The qRT–PCR results were obtained from three biological and three technical replicates, with five samples per biological replicate. Values are means ± SD. The capital letters indicate significant differences in expression among different tissues based on LSD (least significant difference) multiple comparisons (*p* < 0.01).

**Figure 6 ijms-23-13952-f006:**
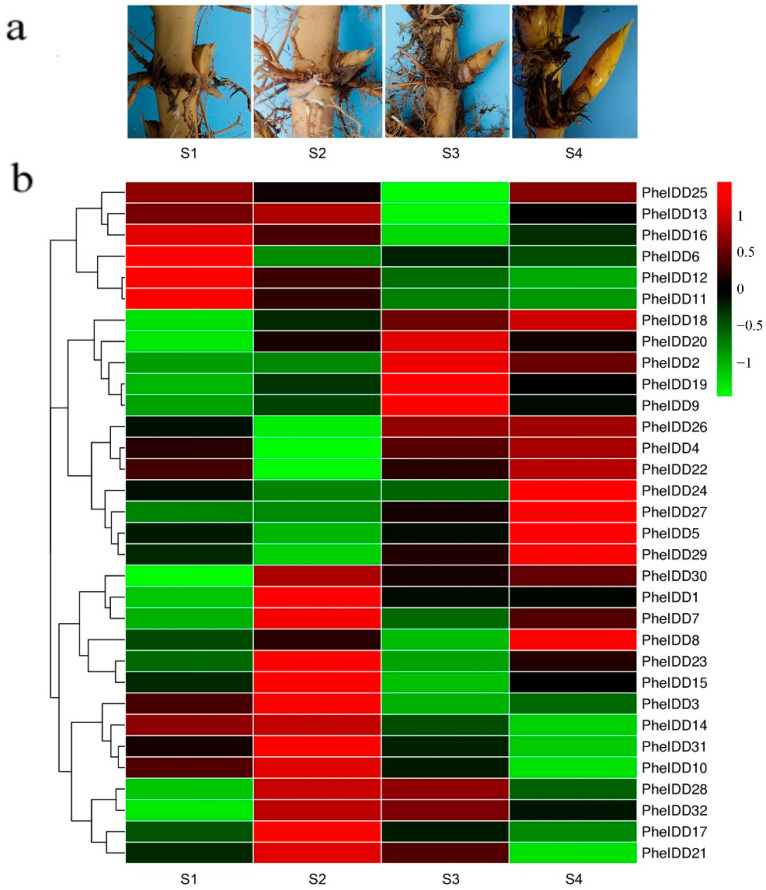
Expression patterns of *PheIDD* family genes in underground buds/shoots at different developmental stages. (**a**) Buds/shoots at different developmental stages on the rhizome. S1: 0.2-cm bud (dormant bud); S2: 2-cm bud (awakening bud); S3: 8-cm shoot; and S4: 16-cm shoot. (**b**) Expression heatmap of *PheIDD* genes in underground buds/shoots at different developmental stages. The color scale indicates the relative expression level. Green in color scale erected vertically at the right side of the picture indicates lower transcript abundance compared to the relevant control, and red indicates higher transcript abundance. The *PheNTB* gene was used as a reference. The qRT–PCR results were obtained from three biological and three technical replicates.

**Figure 7 ijms-23-13952-f007:**
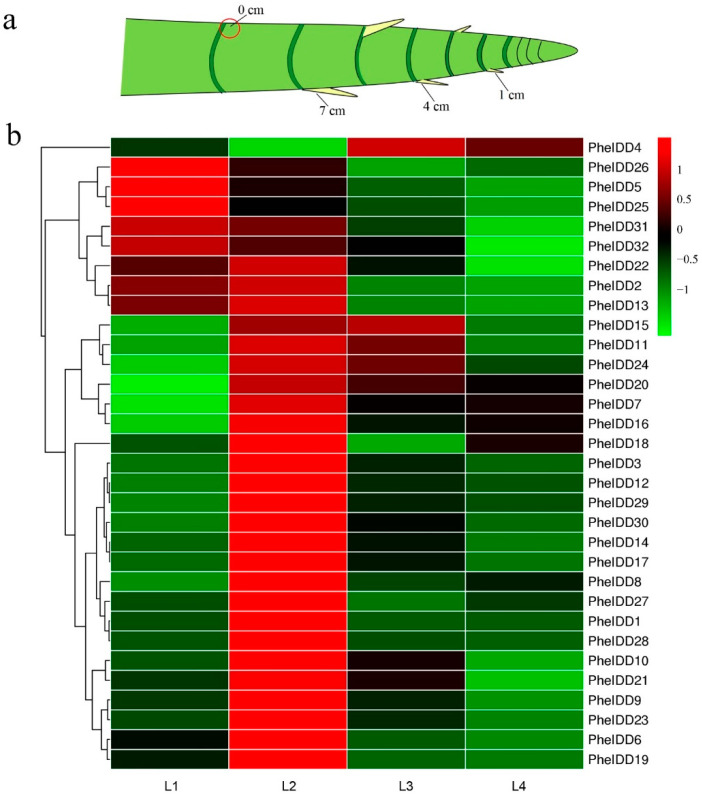
Expression patterns of *PheIDD* family genes in aboveground branches of different sizes. (**a**) Schematic diagram of aboveground branches. (**b**) Expression heatmap of *PheIDD* genes in aboveground branches of different sizes. The color scale indicates the relative expression level. Green in color scale erected vertically at the right side of the picture indicates lower transcript abundance compared to the relevant control, and red indicates higher transcriptabundance. L1: 0-cm bud; L2: 1-cm branch; L3: 4-cm branch; L4: 7-cm branch. The *PheNTB* gene was used as a reference. The qRT–PCR results were obtained from three biological and three technical replicates.

**Figure 8 ijms-23-13952-f008:**
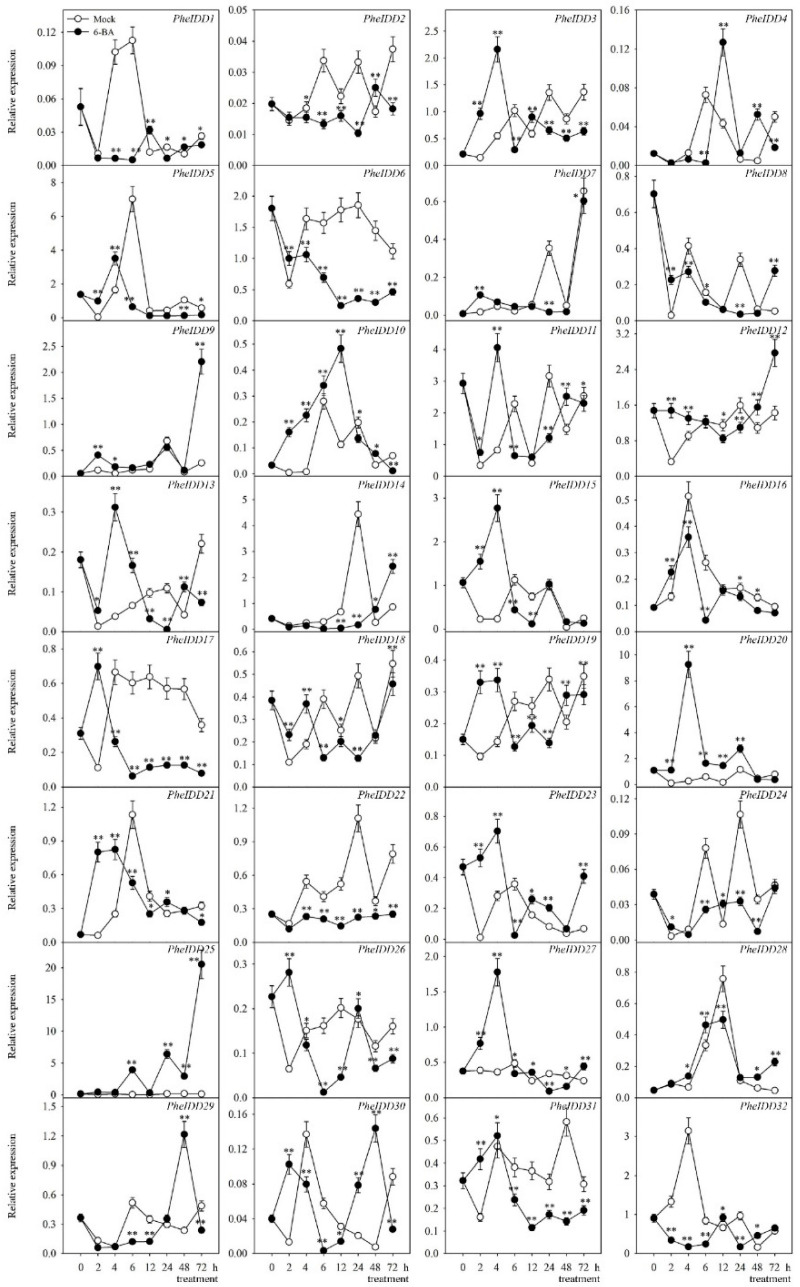
Expression patterns of *PheIDD* family genes under 6-BA treatment. Samples were harvested at 0, 2, 4, 6, 12, 24, 48, and 72 h after treatment. The *PheNTB* gene was used as a reference. The qRT–PCR results were obtained from three biological and three technical replicates. Five 0.6-cm sections of the stem base were harvested per replicate. Values are means ± SD. At a given time point, asterisks indicate a significant difference in expression between 6-BA treatment and Mock treatment based on LSD multiple comparisons. * *p* < 0.05; ** *p* < 0.01.

**Figure 9 ijms-23-13952-f009:**
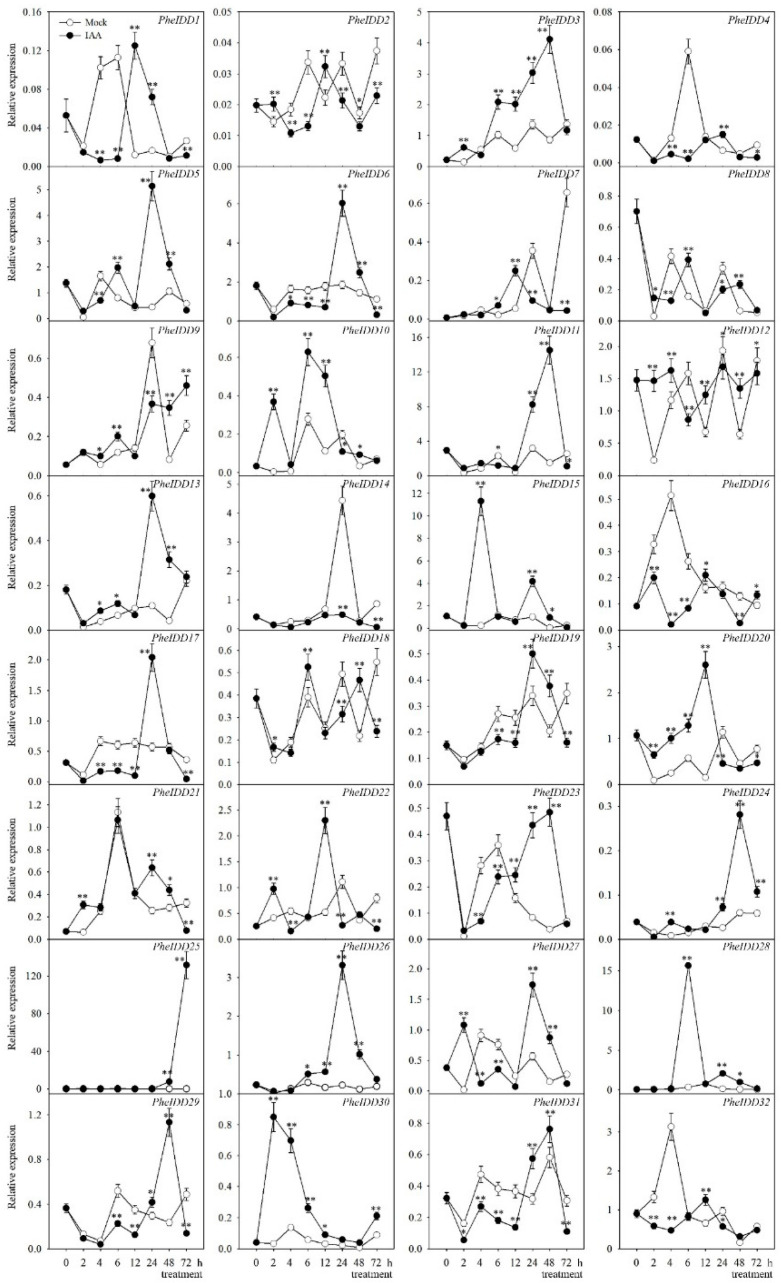
Expression patterns of *PheIDD* family genes under IAA treatment. Samples were harvested at 0, 2, 4, 6, 12, 24, 48, and 72 h after treatment. The *PheNTB* gene was used as a reference. The qRT–PCR results were obtained from three biological and three technical replicates. Five 0.6-cm sections of the stem base were harvested per replicate. Values are means ± SD. At a given time point, asterisks indicate a significant difference in expression between IAA treatment and Mock treatment based on LSD multiple comparisons. * *p* < 0.05; ** *p* < 0.01.

**Figure 10 ijms-23-13952-f010:**
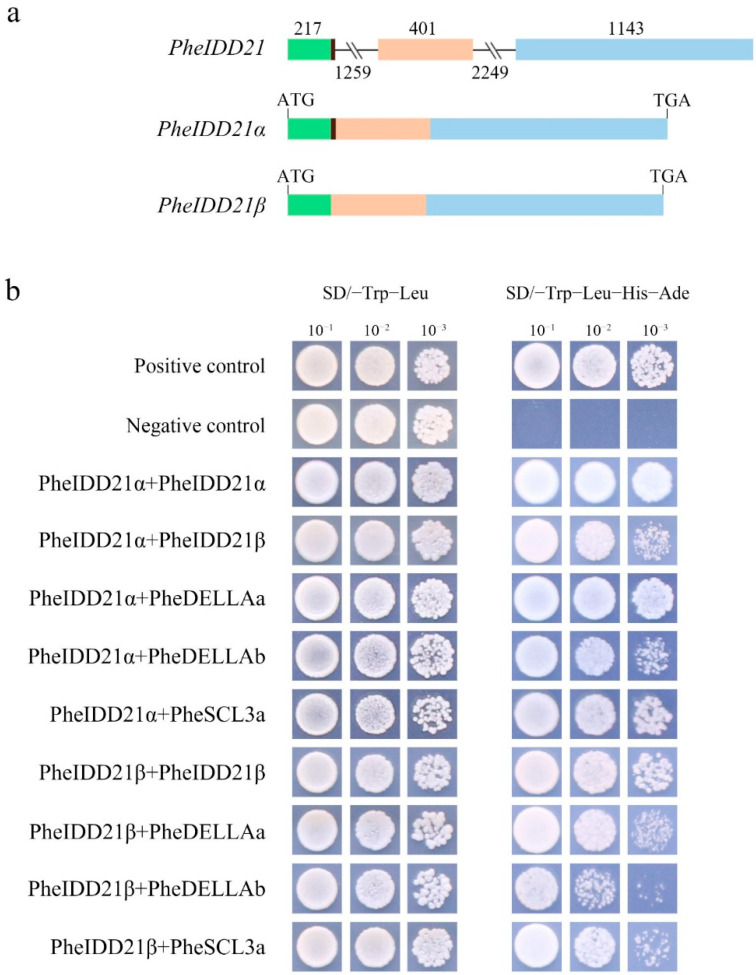
Genomic structures of *PheIDD21* splice variants (**a**) and Y2H assays for interacting proteins of both isoforms (**b**). Lines indicate introns, and boxes of different colors indicate exons. The numbers above the boxes indicate the exon lengths, and the numbers below the lines indicate the intron lengths. ATG indicates the start codon, and TGA indicates the stop codon.

**Figure 11 ijms-23-13952-f011:**
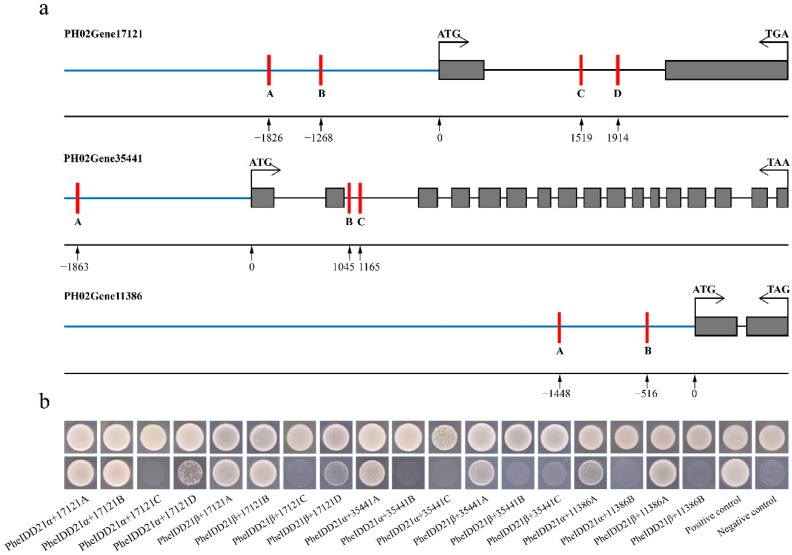
PheIDD21 directly binds to three genes (PH02Gene17121, PH02Gene35441, and PH02Gene11386). (**a**) Schematic diagram of putative *cis*-elements in the three genes, which were used for PheIDD21 binding assays. The capital letters A–D indicate the locations of the *cis*-elements. (**b**) Yeast one-hybrid assays showed binding of PheIDD21α/β to the putative *cis*-elements.

**Table 1 ijms-23-13952-t001:** Paralogous (Phe-Phe) and orthologous (Phe-Os) *IDD* gene pairs in moso bamboo and rice.

Phe-Phe	Phe-Os
PheIDD1/PheIDD2	PheIDD1/OsID1
PheIDD5/PheIDD6	PheIDD2/OsID1
PheIDD7/PheIDD16	PheIDD4/OsIDD7
PheIDD9/PheIDD22	PheIDD5/OsIDD11
PheIDD10/PheIDD14	PheIDD6/OsIDD11
PheIDD11/PheIDD12	PheIDD7/OsIDD8
PheIDD13/PheIDD18	PheIDD8/OsIDD9
PheIDD15/PheIDD17	PheIDD9/OsIDD2
PheIDD19/PheIDD26	PheIDD11/OsIDD1
PheIDD20/PheIDD23	PheIDD12/OsIDD1
PheIDD21/PheIDD25	PheIDD13/OsIDD3
PheIDD27/PheIDD29	PheIDD15/OsIDD6
PheIDD28/PheIDD30	PheIDD16/OsIDD8
PheIDD31/PheIDD32	PheIDD17/OsIDD6
	PheIDD18/OsIDD3
	PheIDD19OsIDD4
	PheIDD20/OsIDD14
	PheIDD21/OsIDD10
	PheIDD22/OsIDD2
	PheIDD23/OsIDD14
	PheIDD24/OsIDD7
	PheIDD25/OsIDD10
	PheIDD26/OsIDD4
	PheIDD27/OsIDD13
	PheIDD28/OsIDD12
	PheIDD29/OsIDD13
	PheIDD30/OsIDD12
	PheIDD31/OsIDD5
	PheIDD32//OsIDD5

## Data Availability

Not applicable.

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
