# Peer review of "Genome-Wide Identification of the Highly Conserved INDETERMINATE DOMAIN (IDD) Zinc Finger Gene Family in Moso Bamboo (Phyllostachys edulis)"

_ijms, 2022, doi:10.3390/ijms232213952_

Round 1

Reviewer 1 Report

This manuscript reports the identification of the INDETERMINATE DOMAIN (IDD) gene family in Moso bamboos. In addition, the expression of IDDs in various tissues and in response to plant hormones were detected. More interesting, the alternative splicing was presented in PheIDD21 gene and generated two alleles, which encoding protein can both interacted with PheDELLA and PheSCL3 and bind to the same downstream target genes. The authors present a great deal of data for exploring the molecular breeding mechanism of lateral organ development in moso bamboo. The manuscript is clearly written, and the experimental approaches are generally appropriate. Nevertheless, there are few points of criticism:

1. The legend of Figure 1 has (a) and (b) panels. But there was no ‘a’ ‘b’ icons in the Figure 1. Please add them.

2. Line 190: ‘root’ tissue is missing from the list of tissues in the description of ‘Tissue specific expression pattern’

3. The statistical analysis of capital letters in Figure 5 is incorrect. For example, the differential expression of PheIDD5 gene in young leaves and tillers is wrong. Please check it.

4. In Figures 8, 9, on the horizontal axis of the graph, h should be deleted from 0-48 and presented only at 72 with one space. Meanwhile, add “treatment” on the horizontal axis.

Author Response

Comments and Suggestions for Authors

This manuscript reports the identification of the INDETERMINATE DOMAIN (IDD) gene family in Moso bamboos. In addition, the expression of IDDs in various tissues and in response to plant hormones were detected. More interesting, the alternative splicing was presented in PheIDD21 gene and generated two alleles, which encoding protein can both interacted with PheDELLA and PheSCL3 and bind to the same downstream target genes. The authors present a great deal of data for exploring the molecular breeding mechanism of lateral organ development in moso bamboo. The manuscript is clearly written, and the experimental approaches are generally appropriate. Nevertheless, there are few points of criticism:

  1. The legend of Figure 1 has (a) and (b) panels. But there was no ‘a’ ‘b’ icons in the Figure 1. Please add them.

Response:Thank you. We have added a and b in Fig.1  

  1. Line 190: ‘root’ tissue is missing from the list of tissues in the description of ‘Tissue specific expression pattern’

Response:Thank you. We have added “root” in the list of tissues

  1. The statistical analysis of capital letters in Figure 5 is incorrect. For example, the differential expression of PheIDD5gene in young leaves and tillers is wrong. Please check it.

Response:Thank you. We have checked all the statistical analysis and revised them in the manuscript.  

  1. In Figures 8, 9, on the horizontal axis of the graph, h should be deleted from 0-48 and presented only at 72 with one space. Meanwhile, add “treatment” on the horizontal axis.

Response:Thank you. We have revised them.

Reviewer 2 Report

The manuscript describes a study identifying IDD family genes in moso bamboo and analyzing the expressions of IDD family genes during development and in response to hormone treatments, exploring the characterization of PheIDD21. They proposed that IDD family genes may be involved in the development of lateral buds. After consideration, we feel it has potential interests for t the general reader of International Journal of Molecular Sciences. However, current version needs following minor revision.

1. Figure1 a and b should also be marked in the Figure1 since the legend has been divided into (a) and (b). 

2. Please marked S1, S2, S3, and S4 in Figure 6a

3. The author should provide more legends for Fig. 6b and Fig. 7b.

4. Line 153 to 154: The scientific finding should be mentioned here.

5. Line 307 Please provide the type of AS (for example, Alternative selection of 5′ or 3′ splice sites)

6. Line309/347 Space separation should be added in ‘PheIDD21αand’

7. Symbols for genes should be italicized. For example, Line no.404 to 405 should be revised.

8. The software version should be added in method section, such as TBtools (Line558).

9. Table S1/Table S2 and all supplemental figures could not be downloaded from the review system.

Author Response

Comments and Suggestions for Authors

The manuscript describes a study identifying IDD family genes in moso bamboo and analyzing the expressions of IDD family genes during development and in response to hormone treatments, exploring the characterization of PheIDD21. They proposed that IDD family genes may be involved in the development of lateral buds. After consideration, we feel it has potential interests for t the general reader of International Journal of Molecular Sciences. However, current version needs following minor revision.

  1. Figure1 a and b should also be marked in the Figure1 since the legend has been divided into (a) and (b). 

Response:Thank you. We have added a and b in Fig.1.

  1. Please marked S1, S2, S3, and S4 in Figure 6a

Response:Thank you. We have marked them in Figure 6a  

  1. The author should provide more legends for Fig. 6b and Fig. 7b.

Response:Thank you. We have provided more for Fig. 6b and Fig. 7b.

  1. Line 153 to 154: The scientific finding should be mentioned here.

Response:Thank you. We have provided the finding here.

  1. Line 307 Please provide the type of AS (for example, Alternative selection of 5′ or 3′ splice sites)

   Response:Thank you. We have provided the information here.

  1. Line309/347 Space separation should be added in ‘PheIDD21αand’

   Response:Thank you. We have added space separation here.

  1. Symbols for genes should be italicized. For example, Line no.404 to 405 should be revised.

Response:Thank you. We have checked all symbols for genes and proteins in the manuscript.

  1. The software version should be added in method section, such as TBtools (Line558).

Response:Thank you. We have added it in method section.

  1. Table S1/Table S2 and all supplemental figures could not be downloaded from the review system.

Response:Thank you. Table S1/Table S2 are in the manuscript and uploaded. I don’t know they are missing in the Manuscript for Revisions. We have uploaded them now.

Reviewer 3 Report

In this study, authors performed genome-wide identification of INDETERMINATE DOMAIN (IDD) zinc finger gene family in Moso bamboo by using in silico tools and they validated their results with qRT-PCR. First of all, I would like to thank all authors for their extensive and valuable study. I have some minor concern about the study:

1. For phylogenetic analysis, why did you use whole gene sequence and why not just domain sequences?

2. Please discuss why there is no ortholog with Arabidopsis?

3. Please discuss what is the main factor determining IDD gene number in the different organism.

Author Response

Comments and Suggestions for Authors

In this study, authors performed genome-wide identification of INDETERMINATE DOMAIN (IDD) zinc finger gene family in Moso bamboo by using in silico tools and they validated their results with qRT-PCR. First of all, I would like to thank all authors for their extensive and valuable study. I have some minor concern about the study:

  1. For phylogenetic analysis, why did you use whole gene sequence and why not just domain sequences?

Response:Thank you. Based on the sequence characteristics of IDD family members, the domain sequences exist at the N-terminal and C-terminal. Therefore, we use whole gene sequences to perform phylogenetic analysis.

  1. Please discuss why there is no ortholog with Arabidopsis?

Response:Thank you. Based on the reference [2], although monocot and dicot genomes have a similar number of IDD genes, monocots present a greater number of sequences clusters, most of them grass-specific. Therefore, it is likely the whole-genome duplication (WGD) gave rise to grass-specific IDD. So, there is no OsID1 ortholog in Arabidopsis

  1. Please discuss what is the main factor determining IDD gene number in the different organism.

     Response:Thank you. Based on the reference [2], WGD may have given rise to monocots having a higher number of IDD clusters than dicots. In addition, additional duplication, especially segmental duplication events have occurred in the IDD family during bamboo evolution.

Reviewer 4 Report

In this paper, Guo et al. systematically analyzed the 32 IDD genes in moso bamboo and examined their expression patterns. It is meaningful to characterize these IDD genes. Nevertheless, this manuscript is largely descriptive, which only provids some basic information and lacks necessary impressive discoveries.

Major concerns:

1. PheIDD1 to PheIDD32 was named based on their homology to OsID1, but in the alignment of amino acid sequences, ZmID1 was used, please explain.

2. All the supplemental figures were missing.

3. Why only examined the responses of PheIDDs to IAA and 6-BA, what about other phytohormones, such as GA and BR?

4. Y2H is not solid enough to show interactions, other evidence should be provided.

5. Lacking functional examination of any PheIDD. The quality of this paper will be highly improved even if only one PheIDD is preliminarilly tested in Arabidopsis or rice.

6. The full text needs a more logical organization and proofreading.

Minor concerns:

1. Personally, delete and alternative splicing of PheIDD21 in the title.

2. In line 54, a space is necessary between AtIDD14α” and accumulates.

3. Line 69, Dehydration should be dehydration.

4. Line 74, binding to the core cis-elements....(cis should be italic).

5. In Figure 1, which is (b)?

6. In Figure 5, 8 and 9, all PheIDDs should be italic.

7. Line 285, P <0.01, P should be itallic.

8. Line 551, should be Analysis of chemical characteristics.

9. In Table S4, Datebase should be Database.

Author Response

Comments and Suggestions for Authors

In this paper, Guo et al. systematically analyzed the 32 IDD genes in moso bamboo and examined their expression patterns. It is meaningful to characterize these IDD genes. Nevertheless, this manuscript is largely descriptive, which only provids some basic information and lacks necessary impressive discoveries.

Major concerns:

  1. PheIDD1 to PheIDD32 was named based on their homology to OsID1, but in the alignment of amino acid sequences, ZmID1 was used, please explain.

Response:Thank you. We named 32 IDD members based on the identity with ZmID1, because the IDD gene was first cloned in maize. OsID1 in rice is also named based on ZmID1. Therefore, we used ZmID1 in the alignment of amino acid sequences. We made a mistake in the manuscript and have now revised it.

  1. All the supplemental figures were missing.

Response:Sorry. The supplemental figures are embedded in the manuscript and uploaded. I don’t know they are missing in the Manuscript for Revisions. We have uploaded them now.

  1. Why only examined the responses of PheIDDs to IAA and 6-BA, what about other phytohormones, such as GA and BR?

Response:Thank you. Moso bamboo is very difficult to flower and the seeds are not easy to obtain. It is more difficult to obtain seeds on a single bamboo plant. Therefore, the seedling materials are not enough. So, we only examined the responses of PheIDDs to IAA and 6-BA.

  1. Y2H is not solid enough to show interactions, other evidence should be provided.

Response:Thank you. Due to the impact of the COVID-19, we can't go back to the laboratory to perform other work within ten days.

  1. Lacking functional examination of any PheIDD. The quality of this paper will be highly improved even if only one PheIDD is preliminarilly tested in Arabidopsis or rice.

Response:Thank you. We did functional study on one gene in Arabidopsis. However, due to the COVID-19, we were unable to return to the laboratory to observe the phenotype and didn’t receive T1 transgenic seeds.

  1. The full text needs a more logical organization and proofreading.

 Response:We have asked the professionals and native English speaker to perform organization and proofreading

Minor concerns:

  1. Personally, delete “and alternative splicing of PheIDD21” in the title.
  2. In line 54, a space is necessary between “AtIDD14α” and “accumulates”.
  3. Line 69, Dehydration should be dehydration.
  4. Line 74, binding to “the” core cis-elements....(cis should be italic).
  5. In Figure 1, which is (b)?
  6. In Figure 5, 8 and 9, all PheIDDs should be italic.
  7. Line 285, P <0.01, “P” should be itallic.
  8. Line 551, should be “Analysis of chemical characteristics”.
  9. In Table S4, Datebase should be “Database”.

  Response:Thank you. We have revised all the minor concerns above.

Round 2

Reviewer 4 Report

Frankly, there is no essential improvement in this revised version. At the current status, this manuscript fails to meet the criteria for publication in IJMS.